# Lipid Process Markers of Durum Wheat Debranning Fractions

**DOI:** 10.3390/foods12163036

**Published:** 2023-08-12

**Authors:** Silvia Marzocchi, Maria Cristina Messia, Emanuele Marconi, Maria Fiorenza Caboni, Federica Pasini

**Affiliations:** 1Department of Agricultural and Food Sciences, University of Bologna, Piazza Goidanich, 60, 47521 Cesena, Italy; silvia.marzocchi4@unibo.it (S.M.); maria.caboni@unibo.it (M.F.C.); 2Department of Agricultural, Environmental and Food Sciences, University of Molise, Via F. De Sanctis, 86100 Campobasso, Italy; messia@unimol.it; 3Research Unit of Food Science and Human Nutrition, Department of Science and Technology for Human and Environment, University Campus Bio-Medico of Rome, Via Álvaro del Pontillo, 21, 00128 Rome, Italy; e.marconi@unicampus.it; 4Interdepartmental Centre of Industrial Agri-Food Research (CIRI Agroalimentare), University of Bologna, Via Quinto Bucci, 336, 47521 Cesena, Italy

**Keywords:** durum wheat, debranning, lipid composition, biomarkers

## Abstract

At present, whole grains are usually obtained by adding bran and middlings to refined flours, and this recombination leads to certain variations in the ratio of endosperm, bran and germ, resulting in flours with very different compositional characteristics and rheological properties. Therefore, this study focuses on the identification of specific lipid markers in different debranning fractions of Italian and Canadian durum wheat blends. The by-products obtained from five different debranning levels (3, 6, 9, 12 and 15%) had a high content of monounsaturated fatty acids and a higher concentration of tocopherols and sterols than the corresponding debranned grains. The Italian and Canadian durum wheat samples did not show significant differences in the content of these bioactive lipid compounds. In particular, palmitic acid, oleic acid, tocopherol isomers and total sterols could be useful biomarkers for evaluating the grain-to-tissue ratio in recombined flours.

## 1. Introduction

Durum wheat (*Triticum turgidum* L. subsp. *durum*) is widely cultivated worldwide, and it is an essential food source that plays an important role in the human diet, providing carbohydrates, proteins and energy. For this reason, it is milled to produce many staple foods. The traditional milling process of wheat grains produces refined flour by separating endosperm of the kernels from the bran and germ fractions. The separation of aleurone cells from the outer bran layers is very difficult; for this reason, the term bran usually includes also the aleurone layer, which is removed with the bran during traditional milling. In the 1990s, the debranning process was introduced upstream of the traditional milling process and its beneficial effects are now well known. Debranning involves the gradual removal of the outer skin from the surface of the inward kernels through friction or abrasion. This means that the aleurone layer could remain attached to the endosperm and the outermost layers, which could contain contaminants and microorganism, are removed [1,2]. The debranning process is becoming increasingly popular in the milling and bakery industries, and several studies report its positive effects on flour quality compared to conventional milling, including an improved final flour colour [1,3], an increase of more than 8% in ash, a decrease of about 20% in α-amylase activity, and a decrease of 71% of the deoxynivalenol levels [1,4]. Posner and Hoseney [5] also reported an increase in milling loads due to a reduction in different necessary stages in a milling flow sheet. Debranning also produces new valuable by-products and novel cereal-based foods with the use of debranned cereal kernels. Sun and co-authors [6] observed a positive effect of wheat debranning on the quality of bread after in vitro digestion, with a degree of debranning between 3% and 7% offering the highest nutritional value.

Flours currently on the market are obtained via a milling process that aims to extract/separate endosperm from wheat caryopsis to obtain refined flour or semolina. The other parts of the caryopsis (rich in fiber, vitamins and minerals) are considered by-products, including mainly bran and fine bran. In Italy, products obtained from durum wheat that are intended for human consumption are regulated by the Italian Presidential Decree No. 187/2001, which classifies flour according to the degree of refining, as expressed by the ash content. Currently, in most cases, wholemeal type flours are obtained by recombining appropriate amounts of bran and middlings with refined flours. However, this recombination results in products with very different compositional characteristics and technological properties, which also affects the organoleptic profiles of whole-grain end products. This is the consequence of the lack of a clear and precise definition of whole-grain flour in the Italian Presidential Decree No. 187/2001. Moreover, the European Food Safety Authority, in a whole-grain-related health claim opinion [7], provides the definition of the American Association of Cereal Chemists which states that “whole grain consists of the intact, ground, cracked or flaked caryopsis, whose principal anatomical components—the starchy endosperm, germ and bran–are present in the same relative proportions as they exist in the intact caryopsis”. In processed whole-wheat products, this cannot be traceably and unambiguously transferred due to a lack of markers and biomarkers of different kernel fractions, such as germ, the aleurone layer and bran. Consequently, recombination per grain and variety leads to certain variations in the ratio of endosperm, bran and germ between flour and product lots, which have very different compositional and rheological characteristics.

For these reasons, in this study, the lipid components of the fractions obtained from different percentages of debranning of durum wheat kernels were characterized. In particular, qualitative and quantitative analyses of fatty acids, tocochromanols and phytosterols were performed in debranned grains and in debranning by-products to identify specific lipid markers.

## 2. Materials and Methods

### 2.1. Chemicals and Reagents

All reagents were purchased from Merck (Darmstadt, Germany). GLC-463 mix was from Nu-Check (Elysian, MN, USA), while FAME 189-19, (+)-α-tocopherol, 5α-cholestan-3β-ol, campesterol, campestanol, stigmasterol, β-sitosterol, sitostanol and Δ^5^-avenasterol were from Sigma-Aldrich (St. Louis, MO, USA).

### 2.2. Samples

Italian (I) and Canadian (C) blends of durum wheat (*Triticum turgidum* L. subsp. *durum*), which were obtained from a local distributor, were subjected to five different levels (3, 6, 9, 12 and 15%) of a debranning process, using a lab-scale debranner (TakaYama, Taichung, Taiwan), thereby recovering debranned durum wheat kernels (DG) and debranning by-products (DB). A summary of the samples is presented in Table 1.

### 2.3. Lipid Extraction

Oil was extracted using a Soxhlet apparatus (Behr Labor-Technik, Fischer Scientific Italia, Milano, Italy), as reported by Marzocchi et al. [8] and in accordance with the AOAC Official Method [9] (AOAC International Website). About 10 g of ground wheat fractions were placed in a cellulose thimble, and oil was extracted using a refluxing *n*˗hexane. After using a rotary evaporator to remove the solvent, the oil was taken up with *n*-hexane/isopropanol (4:1 *v*/*v*) solution and stored at −18 °C until use. The oil content of the wheat fractions was expressed as percentage (%) on the total fresh weight of each sample. Each extraction was performed two times (*n* = 2).

### 2.4. Fatty Acid Analysis

The fatty acid composition from the oil of the wheat fractions was determined as fatty acid methyl esters (FAMEs) via capillary gas chromatography (CGC) analysis after alkaline treatment under the conditions reported by Marzocchi et al. [10]. FAMEs were analyzed using a GC 2010 Plus gas chromatograph equipped with a flame ionization detector (FID) and an AOC-20s auto sampler (Shimadzu Corporation, Kyoto, Japan) and using a fused silica capillary column BPX70 (10 m × 0.1 mm i.d., 0.2 μm f.t.) from SGE Analytical Science (Ringwood, VIC, Australia). FAMEs were identified by comparing the peak retention time with GLC-463 and FAME 189-19 standard mixtures, and they were quantified by comparing the peak area of each compound with that of the internal standard (C13:0, 2 mg/mL). The FAME composition was expressed as weight percentage of total FAME (mg/100 mg of FAME), and two replicates for each lipid extract (*n* = 4) were analyzed.

### 2.5. Tocochromanol Analysis

Tocochromanols were determined using an HPLC (Agilent 1200 series, Palo Alto, CA, USA) equipped with a fluorimetric detector (FLD) as previously described by Ben Lajnef et al. [11]. A total of 50 mg of oil was dissolved in 0.5 mL of *n*-hexane, and then the extracts were filtered using a 0.45 µm nylon filter. The chromatographic separation was carried out using a HILIC Poroshell 120 column (100 mm × 3 mm and 2.7 μm particle size; Agilent Technologies, Palo Alto, CA, USA) under isocratic conditions with *n*-hexane/ethyl acetate/acetic acid (97.3:1.8:0.9 *v*/*v*/*v*) as the mobile phase and with a flow rate of 0.8 mL/min.

The identification of tocochromanols was conducted by setting the excitation wavelength at 290 nm and the emission wavelength at 325 nm. Tocopherols were identified via co-elution with the standards, whereas tocotrienols were identified by comparing their retention times with that of tocotrienols in a barley extract obtained via hot saponification. α-Tocopherol standard solutions (from 1 to 100 μg/mL) were used for calibration curve construction for quantification. Two replicates for each lipid extract (*n* = 4) were performed.

### 2.6. Sterol Analysis

The procedure for sterol analysis was the same as reported by Marzocchi et al. [8]. After the addition of 0.5 mL of 5α-cholestan-3β-ol (c = 2 mg/mL) as the internal standard, 250 mg of oil were saponified with 10 mL of 2 N methanolic potassium hydroxide at room temperature for 20 h [12]. Subsequently, the organic fraction was washed with 10 mL of diethyl ether and 10 mL of water. The unsaponifiable fraction was further extracted twice with 10 mL of diethyl ether, 10 mL of 0.5 N aqueous KOH and 10 mL of distilled water, respectively. The organic solvent was removed under vacuum, and the unsaponifiable fraction was silylated [13] and used for sterol analysis via GC/MS (GCMS-QP2010 Plus, Shimadzu, Tokyo, Japan) under the same chromatographic conditions as reported by Cardenia et al. [14]. The identification of sterols was carried out by comparing mass spectra and retention time with those of the corresponding chemical standards and by comparing them to the GC/MS data reported in the literature [15]. The concentration of each individual sterol was obtained via the use of the internal standard and expressed as mg/100 g of fat. The analysis was conducted in 2 replicates for each lipid extract (*n* = 4).

### 2.7. Statistical Analysis

The Statistica 8 software (2006, StatSoft, Tulsa, OK, USA) was used to calculate one-way ANOVA (analysis of variance) coupled with Tukey’s honest significant difference (HSD) test. *p*-Values lower than 0.05 were considered statistically significant. All results were expressed as the averages of two replicates for each extract (*n* = 4 for each sample), and the analytical data were used for statistical comparisons.

## 3. Results and Discussion

### 3.1. Lipid Content

By removing individual bran layers from the kernels, the debranning process produces different levels of by-products with a significantly higher fat content than the remaining debranned kernels, because of the different chemical composition of inner and outer layers of caryopsis. Indeed, the results in Figure 1 show how debranning at 6, 9, 12 and 15% produced by-products with the highest fat content at a significant level (*p* < 0.05), from 6.3 (I-DB6) to 7.4% (C-DB15), compared to the debranning by-products at 3% that showed values equal to 5.4 and 5.3% for Italian and Canadian samples, respectively. All corresponding counterparts, and thus the debranned grains (DG) obtained at the five levels of debranning, presented a significantly (*p* < 0.05) lower lipid percentage compared to the stripped-off outer layers. These values ranged from 0.5 (I-DG15) to 1.6% (C-DG15), and they did not show significant differences between them. The Italian and Canadian wheat grains presented the same trend and lipid content in the different debranning fractions.

The difference in lipid content confirmed the different quantities of aleurone and germ present in the debranning by-product samples. Unlike the debranned grains, the DB3, DB6, DB9, DB12 and DB15 were probably characterized with the presence of germ that is known to contain more oil than the other bran layers [16].

Figure 1 shows the lipid content (%) in different products obtained from the debranning process of Italian (I) and Canadian (C) durum wheat.

### 3.2. Fatty Acids

In the Italian and Canadian samples of durum wheat, the analysis identified and quantified 21 fatty acids (FA), ranging from lauric acid (C12:0) to nervonic acid (C24:1). As reported in Table 2, the most abundant FAs in the fractions obtained via the debranning process were as follows: linoleic acid (C18:2*n6*) > oleic acid (C18:1*c9*) > palmitic acid (C16:0) > linolenic acid (C18:2*n3*) > stearic acid (C18:0). These data are in line with other studies reported in the literature about FAs in durum wheat [16,17]. Other minor FAs were also detected with levels lower than 0.1% (C12:0, C14:0, C15:0, C16:1c, C17:0, C17:1, C18:1c11, C20:0, C20:1, C20:2n6, C22:0, C22:1, C22:2, C22:4, C24:0 and C24:1).

As shown in Figure 2, in all the samples analyzed, PUFAs were present with a percentage ranging from 60 to 64%, and they were characterized by higher levels of linolenic acid (C18:3*n3*) in the debranning by-products compared to the corresponding debranned grains; less defined and clear was the difference in the content of linoleic acid (C18:2*n6*) in the various fractions of decortication. MUFAs (17–21%), represented mainly as oleic acid (C18*c9*), showed significantly higher (*p* < 0.05) levels in the debranning by-product samples. Finally, the highest levels of SFAs (17–20%), consisting principally of palmitic (C16:0) and stearic acid (C18:0), were found in the debranned grain (DG) samples for all the different debranning percentages. As for MUFAs, the SFA results showed the highest level of this class of FA. Italian and Canadian wheat did not show significant variation in the FA content and distribution in the different debranning samples analyzed.

### 3.3. Tocochromanol Determination

In all Italian and Canadian durum wheat decortication fractions, five tocochromanols were identified and quantified: three tocopherols (α-tocopherol, β-tocopherol, and γ-tocopherol) and two tocotrienols (α-tocotrienol and β-tocotrienol) (Appendix A). As shown in Table 3, the various debranning products reported significantly different tocochromanol contents.

The debranning fractions showed β-tocotrienol (49–119 and 36–137 mg/100 g of fat in the Italian and Canadian wheat, respectively) and α-tocopherol (38–87 and 43–68 mg/100 g of fat in the Italian and Canadian wheat, respectively) as the principal ones, followed by α-tocotrienol, β-tocopherol and γ-tocopherol (Table 3). In particular, tocopherol isomers presented a higher content in the by-product samples at all different levels of debranning (DB3, DB6, DB9, DB12 and DB15); in contrast, α- and β-tocotrienols were more concentrated in all the debranned grain (DG) samples. Considering the total tocochromanol content, the Italian and Canadian wheat showed the highest concentrations at a significant level in different fractions after decortication; in fact, the Italian wheat showed the highest (*p* < 0.05) concentration in the DG3 and DG6 samples (246.5 and 245.6, respectively), while the Canadian wheat exhibited the highest concentration in the DG12 sample (265.6 mg/100 g of fat).

Comparing the Italian and Canadian grains, the samples registered the same trend but with more significant (*p* < 0.05) difference in content between debranned grains and debranning by-products in the Canadian samples. Furthermore, the greatest differences (*p* < 0.05) between the debranned grain and debranning by-product samples were recorded for the level of debranning at 15%.

Additionally, the Italian samples generally presented a higher content of tocochromanols compared to the Canadian samples at the 3 and 6% of debranning levels. In contrast, higher levels of tocochromanols were registered for the Canadian fractions compared to the Italian ones at the 9, 12 and 15% of debranning levels.

At all debranning levels, the total tocochromanol content was higher (*p* < 0.05) in the DG fractions compared to the DB ones, and this result is mainly due to the high contribution of α- and β-tocotrienols that are principally present in the endosperm of kernels. Indeed, according to the literature, the fractions with germ (decortication by-products) are usually richer in tocopherols while the fractions with more endosperm (debranned grains) are richer in tocotrienol [18,19,20]. β-Tocotrienol has already been reported in the literature as the main vitamer in dehulled wheats and is known to reduce serum LDL cholesterol through its antioxidant action [19].

Overall, these results are in line with the percentage proportions of tocochromanols in wheat (30% of α-tocopherol, 18% of α-tocotrienol, 7% β-tocopherol and 44% of β-tocotrienol) evaluated by other authors [21].

### 3.4. Sterols

Sterols are known for their several health-promoting effects like lowering of blood cholesterol and anticancer properties. Because of the high amount typically consumed in the human diet, cereals are a good source of sterols [22]. Significant differences in sterol content between the different debranning fractions of the Italian and Canadian durum wheat samples were noted (Table 4). According to the literature [17,21,23], eight phytosterols that are typically found in wheats were identified in all samples: three saturated sterols (campestanol, sitostanol and avenastanol), four Δ^5^-sterols (campesterol, β-sitosterol, avenasterol and stigmasterol) and one Δ^7^-sterol (Δ^7^-avenasterol) (Appendix A).

The grain fractions obtained after different levels of debranning showed β-sitosterol as the principal one (671–878 and 689–1002 mg/100 g of sterols in the Italian and Canadian wheat, respectively), followed by campesterol, campestanol, sitostanol, Δ^5^-avenasterol, stigmasterol, Δ^7^-avenasterol and avenastanol. In general, all debranning by-product samples (DB3, DB6, DB9, DB12 and DB15) presented significantly (*p* < 0.05) higher sterol content compared to the corresponding debranning grain samples (DG3, DG6, DG9, DG12 and DG15), both for the Italian and Canadian grains. This result is in line with the literature, and it is due to the presence of bran and germ in the debranning by-products, which are parts of the caryopsis that is rich in sterols [8,22,24].

When looking at the total sterol content, the DB6 and DB12 fractions of both Italian and Canadian wheat showed the highest concentrations at a significant level (*p* < 0.05): 2175 and 2126.6 mg/100 g of sterols for the Italian wheat, respectively, and 2538 and 2568 mg/100 g of sterols for the Canadian wheat, respectively. In addition, in the Canadian wheat, the DB15 fraction also showed a high concentration of sterols equal to 2633.4 mg/100 g of sterols, which is the highest value registered compared to all other samples. Debranning resulted in a generalized loss of phytosterol, and the highest variation was observed for the Italian grain processed at a 15% of debranning level. In particular, the Italian and Canadian samples showed the same difference between the DG and DB fractions at the levels of 3, 6, 9 and 12% of debranning: the DB3, DB6, DB9 and DB12 presented about 25, 17, 13 and 15% higher sterol content than the corresponding DG3, DG6, DG9 and DG12 samples. On the other hand, the Italian DB15 sample showed a 33% higher content than the DG15 sample, while the Canadian DB15 sample had a total sterol content that was only about 15% higher than the DG sample.

Different studies on wheat sterols [21,25] confirmed the concentrations obtained in this research (38–42% of β-sitosterol, 18–19% of campesterol, 10% of stanols and 3% of stigmasterol), even though there were some slightly different values due to the different genetic and environmental conditions, such as cultivars, locations, soil properties, and climatic conditions and agronomic practices, which could affect the phytosterol composition found in wheat [21,26,27].

## 4. Conclusions

Recombination of milling fractions to obtain whole grains results in products with very different chemical properties and rheological characteristics. The identification of markers capable of distinguishing specific fractions used during the recombination process is undoubtedly useful.

As for lipid components, the concentration of the same fatty acids increased with the debranning degree; palmitic acid can be considered a marker that is detectable in the debranned grains, whereas oleic acid is detectable in the debranning by-products. Tocochromanols can also be considered lipid markers; in particular, α-tocotrienol and β-tocotrienol were found to be more concentrated in debranned grains than the corresponding debranning by-products which were characterized mainly by α, β and γ-tocopherols. In addition, their total concentrations were different in the Italian and Canadian wheat samples, decreasing or increasing with the debranning degree. Campesterol and stigmasterol were highly concentrated in the debranning by-products at a lower debranning degree (3%), while all other sterols were characteristic of the debranned grains at a higher debranning degree.

Assessment of lipid biomarkers in flours obtained via recombination could provide a useful strategy for tracing the fractions used and determining which layers of outer teguments and germ are included. Although additional studies on a larger number of wheat varieties are necessary in order to confirm the importance of lipid biomarkers, this study represents the first step to help obtain whole-grain flours and products with higher homogeneity and quality.

## Figures and Tables

**Figure 1 foods-12-03036-f001:**
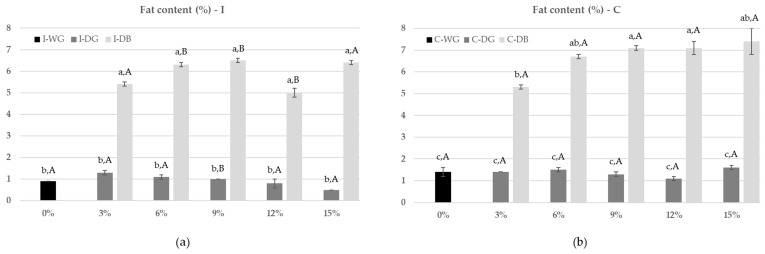
Lipid content (%) in different fractions (DG: debranned grain; DB: debranning by-product) of Italian (**a**) and Canadian (**b**) durum wheat at different levels of debranning (0, 3, 6, 9, 12, and 15%). Data expressed as means ± standard deviation (*n* = 4). Different lowercase letters mean significant differences (*p* < 0.05) among the debranning fractions within the same durum wheat variety (Italian and Canadian); different capital letters mean significant differences (*p* < 0.05) between the same Italian and Canadian durum wheat debranning fraction.

**Figure 2 foods-12-03036-f002:**
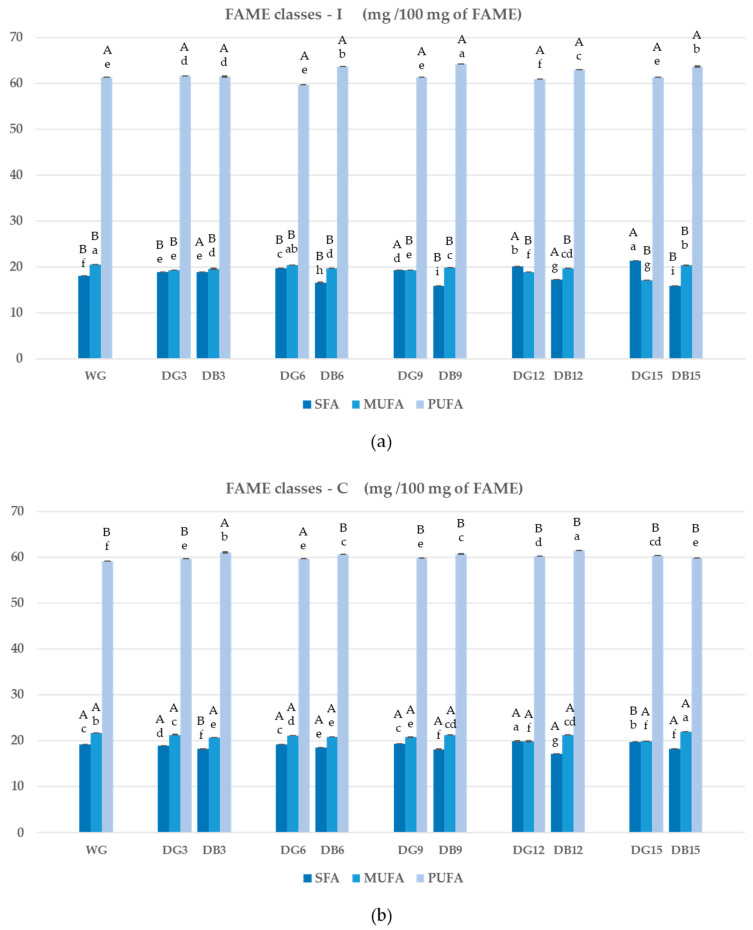
Content of fatty acid classes (mg/100 mg of FAME) in different fractions (DG: debranned grain; DB: debranning by-product) of Italian (**a**) and Canadian (**b**) durum wheat at different levels of debranning (0, 3, 6, 9, 12, and 15%). Abbreviation: SFA: saturated fatty acids; MUFA: monounsaturated fatty acids; PUFA: polyunsaturated fatty acids. Data are expressed as means ± standard deviation (*n* = 4). Different lowercase letters mean significant differences (*p* < 0.05) in the SFA, MUFA and PUFA contents among the debranning fractions within the same durum wheat variety (Italian and Canadian); different capital letters mean significant differences (*p* < 0.05) in the SFA, MUFA and PUFA contents between the same Italian and Canadian durum wheat debranning fraction.

**Table 1 foods-12-03036-t001:** Samples obtained from debranning process of Italian and Canadian durum wheat kernels.

	Italian Durum Wheat	Canadian Durum Wheat
Whole Grain	I-WG	C-WG
3% Debranned Grain	I-DG3	C-DG3
3% Debranning By-product	I-DB3	C-DB3
6% Debranned Grain	I-DG6	C-DG6
6% Debranning By-product	I-DB6	C-DB6
9% Debranned Grain	I-DG9	C-DG9
9% Debranning By-product	I-DB9	C-DB9
12% Debranned Grain	I-DG12	C-DG12
12% Debranning By-product	I-DB12	C-DB12
15% Debranned Grain	I-DG15	C-DG15
15% Debranning By-product	I-DB15	C-DB15

**Table 2 foods-12-03036-t002:** Fatty acid composition and content (mg/100 mg of FAME) of different samples of Italian and Canadian durum wheat at different levels of debranning.

	C16:0	C18:0	C18:1*c9*	C18:2*n6*	C18:3*n3*	Others
I-WG	16.0 ± 0.0 eB	1.2 ± 0.0 eA	18.8 ± 0.1 abB	55.7 ± 0.1 eA	4.9 ± 0.0 dA	3.4 ± 0.1 aA
I-DG3	16.7 ± 0.0 dA	1.6 ± 0.0 dB	17.9 ± 0.0 cdB	56.8 ± 0.0 cA	4.8 ± 0.0 eA	2.3 ± 0.0 bB
I-DB3	16.8 ± 0.0 dA	1.2 ± 0.0 eB	17.9 ± 0.2 cdB	55.6 ± 0.0 eB	5.8 ± 0.0 bA	2.7 ± 0.2 bA
I-DG6	17.1 ± 0.0 cA	2.0 ± 0.0 bA	19.0 ± 0.2 aB	54.7 ± 0.1 fB	4.6 ± 0.0 fA	2.5 ± 0.1 bA
I-DB6	14.9 ± 0.0 gB	0.9 ± 0.0 gB	18.0 ± 0.1 cB	57.6 ± 0.1 bA	5.8 ± 0.0 aA	2.7 ± 0.2 bA
I-DG9	16.8 ± 0.0 dA	1.8 ± 0.0 cB	17.8 ± 0.0 dB	56.4 ± 0.0 dA	4.7 ± 0.0 eA	2.6 ± 0.0 bA
I-DB9	14.3 ± 0.0 hB	0.9 ± 0.0 gB	18.3 ± 0.0 bcB	58.3 ± 0.1 aA	5.8 ± 0.0 abA	2.5 ± 0.0 bA
I-DG12	17.4 ± 0.0 bA	1.8 ± 0.0 cB	17.4 ± 0.2 dB	56.2 ± 0.0 dB	4.6 ± 0.0 fA	2.5 ± 0.1 bA
I-DB12	15.4 ± 0.0 fA	1.1 ± 0.0 fB	18.2 ± 0.1 cB	57.4 ± 0.0 bA	5.4 ± 0.0 cA	2.4 ± 0.1 bA
I-DG15	18.3 ± 0.0 aA	2.2 ± 0.0 aA	15.8 ± 0.2 eB	56.5 ± 0.1 cdB	4.6 ± 0.0 fA	2.5 ± 0.0 bA
I-DB15	14.1 ± 0.0 iB	1.1 ± 0.0 fB	18.8 ± 0.1 abB	58.0 ± 0.1 aA	5.5 ± 0.0 cA	2.5 ± 0.0 bA
C-WG	16.6 ± 0.0 dA	1.8 ± 0.0 cA	20.2 ± 0.2 abA	55.0 ± 0.0 eB	3.8 ± 0.0 eB	2.6 ± 0.2 aB
C-DG3	16.5 ± 0.0 dB	1.8 ± 0.0 cA	19.1 ± 0.1 bcA	55.6 ± 0.0 dB	3.7 ± 0.0 fB	2.5 ± 0.0 abcA
C-DB3	15.9 ± 0.0 fB	1.4 ± 0.1 eA	19.9 ± 0.0 dA	56.1 ± 0.1 cA	4.8 ± 0.0 aB	2.6 ± 0.0 aA
C-DG6	16.6 ± 0.0 dB	1.9 ± 0.0 bB	19.7 ± 0.1 cA	56.0 ± 0.0 cA	3.6 ± 0.0 gA	2.2 ± 0.1 cB
C-DB6	16.2 ± 0.0 eA	1.6 ± 0.0 dA	19.2 ± 0.1 dA	56.0 ± 0.1 cB	4.4 ± 0.0 bB	2.6 ± 0.0 aA
C-DG9	16.7 ± 0.0 cA	2.0 ± 0.0 abA	19.2 ± 0.1 dA	56.1 ± 0.1 cA	3.5 ± 0.0 hB	2.5 ± 0.0 abA
C-DB9	15.9 ± 0.0 fA	1.6 ± 0.0 dA	19.7 ± 0.0 cA	56.2 ± 0.1 cB	4.4 ± 0.0 cB	2.3 ± 0.0 bcA
C-DG12	17.1 ± 0.0 aB	2.0 ± 0.0 aA	18.5 ± 0.1 eA	56.5 ± 0.0 bA	3.4 ± 0.0 iB	2.4 ± 0.1 abcA
C-DB12	14.8 ± 0.0 gB	1.6 ± 0.0 dA	19.8 ± 0.0 bcA	57.0 ± 0.0 aB	4.3 ± 0.1 cB	2.4 ± 0.0 abcA
C-DG15	17.0 ± 0.0 bB	2.0 ± 0.0 aB	18.5 ± 0.1 eA	56.8 ± 0.0 bA	3.4 ± 0.0 iB	2.3 ± 0.0 abcA
C-DB15	15.9 ± 0.0 fA	1.6 ± 0.0 dA	20.4 ± 0.1 aA	55.5 ± 0.0 dB	4.0 ± 0.0 dB	2.5 ± 0.0 abcA

Abbreviation: I: Italian; C: Canadian; DG: debranned grain; DB: debranning by-product; others include C12:0, C14:0, C15:0, C16:1c, C17:0, C17:1, C18:1c11, C20:0, C20:1, C20:2n6, C22:0, C22:1, C22:2, C22:4, C24:0 and C24:1. Data are expressed as means ± standard deviation (*n* = 4). In each column, different lowercase letters mean significant differences (*p* < 0.05) among the fractions of each individual durum wheat variety (Italian and Canadian), whereas different capital letters mean significant differences (*p* < 0.05) between the same Italian and Canadian durum wheat fraction for each fatty acid.

**Table 3 foods-12-03036-t003:** Tocochromanol composition and content (mg/100 mg of fat) in different samples of Italian and Canadian durum wheat at different levels of debranning.

	α-Tocopherol	α-Tocotrienol	β-Tocopherol	γ-Tocopherol	β-Tocotrienol	Total
I-WG	77.3 ± 0.4 abA	36.4 ± 0.3 cdA	14.7 ± 0.7 cA	2.1 ± 0.3 bA	71.1 ± 2.3 cdA	201.6 ± 4.1 bcA
I-DG3	73.1 ± 0.8 bcA	55.0 ± 3.7 aA	21.2 ± 1.7 abA	n.d.	97.2 ± 6.2 bA	246.5 ± 10.8 aA
I-DB3	80.3 ± 7.3 abA	35.5 ± 0.8 cdA	27.2 ± 0.9 aA	4.9 ± 0.8 aA	56.4 ± 0.9 eA	204.3 ± 10.7 bcA
I-DG6	67.3 ± 0.3 cdA	52.1 ± 1.2 aA	16.0 ± 2.0 bA	n.d.	110.2 ± 0.4 aA	245.6 ± 3.3 aA
I-DB6	87.2 ± 2.5 aA	32.2 ± 0.5 dB	20.1 ± 0.3 bA	2.5 ± 0.0 bB	54.9 ± 0.5 eB	196.9 ± 3.8 cA
I-DG9	54.9 ± 0.8 cefA	39.9 ± 0.4 bcB	12.8 ± 0.2 cdA	n.d.	89.9 ± 1.2 bA	197.5 ± 2.6 cB
I-DB9	65.0 ± 3.1 deA	39.4 ± 1.4 bcB	12.1 ± 0.1 deA	2.7 ± 0.0 bB	67.7 ± 1.6 dB	186.9 ± 6.0 cA
I-DG12	38.0 ± 1.8 fB	41.2 ± 1.8 bB	8.2 ± 0.2 efB	n.d.	79.6 ± 0.3 cB	167.0 ± 3.8 dB
I-DB12	61.8 ± 0.4 eA	38.1 ± 0.4 bcB	11.5 ± 0.1 deA	2.3 ± 0.3 bB	68.1 ± 0.1 dB	181.8 ± 0.5 cB
I-DG15	45.5 ± 2.3 fA	42.9 ± 1.7 bA	14.4 ± 1.6 cA	n.d.	118.5 ± 2.2 aA	221.3 ± 7.9 bA
I-DB15	67.2 ± 0.2 cdA	33.5 ± 0.2 eA	14.5 ± 0.3 cB	n.d.	49.0 ± 0.2 fB	164.2 ± 0.2 dB
C-WG	56.7 ± 1.0 cdeB	26.6 ± 0.4 fB	12.2 ± 0.2 cB	1.8 ± 0.1 dA	50.9 ± 1.2 fB	148.2 ± 0.4 eB
C-DG3	53.8 ± 2.0 deB	38.2 ± 1.1 deB	11.0 ± 0.8 dB	n.d.	77.2 ± 0.3 dB	180.2 ± 4.2 dB
C-DB3	67.5 ± 0.5 aB	28.8 ± 0.5 fB	11.4 ± 0.1 dB	3.0 ± 0.2 cB	35.5 ± 0.6 gB	146.2 ± 1.9 eB
C-DG6	50.7 ± 2.0 eB	48.6 ± 0.6 bB	14.7 ± 0.0 bcB	n.d.	76.8 ± 0.4 dB	190.8 ± 3.0 cB
C-DB6	64.8 ± 0.3 bB	43.4 ± 0.6 cdeA	14.6 ± 1.6 bB	6.6 ± 1.1 aA	67.2 ± 0.7 eA	196.6 ± 0.2 cA
C-DG9	56.1 ± 0.6 cdeA	45.8 ± 1.0 bcA	12.0 ± 0.2 bcA	n.d.	94.2 ± 2.2 cA	208.1 ± 0.8 bA
C-DB9	62.0 ± 0.2 bcA	45.0 ± 2.1 bcA	12.5 ± 0.4 cA	4.3 ± 0.1 bA	73.3 ± 0.3 dA	197.1 ± 2.8 cA
C-DG12	58.1 ± 2.3 cdA	58.5 ± 0.6 aA	12.0 ± 0.7 cdA	n.d.	137.0 ± 1.3 bA	265.6 ± 1.0 aA
C-DB12	63.6 ± 1.3 bA	46.6 ± 0.2 bA	11.4 ± 0.4 dA	4.7 ± 0.2 bA	76.4 ± 1.4 dA	202.7 ± 0.6 cA
C-DG15	42.7 ± 2.3 fA	40.3 ± 0.9 dA	11.4 ± 0.4 dB	n.d.	107.8 ± 1.8 aB	202.2 ± 3.9 cB
C-DB15	63.1 ± 2.1 bA	35.3 ± 1.1 eA	18.4 ± 1.6 aA	6.9 ± 0.9 aA	74.8 ± 1.1 dA	198.5 ± 2.7 cA

Abbreviation: I: Italian; C: Canadian; DG: debranned grain; DB: debranning by-product. Data are expressed as means ± standard deviation (*n* = 4). In each column, different lowercase letters mean significant differences (*p* < 0.05) among the fractions of each individual durum wheat variety (Italian and Canadian), whereas different capital letters mean significant differences (*p* < 0.05) between the same Italian and Canadian durum wheat fraction for each tocochromanol.

**Table 4 foods-12-03036-t004:** Sterol composition and content (mg/100 g of fat) in different samples of Italian and Canadian durum wheat after the debranning process.

	Campesterol	Campestanol	Stigmasterol	β-Sitosterol	Sitostanol	Δ^5^-Avenasterol	Avenastanol	Δ^7^-Avenasterol	Total
I-WG	358.7 ± 1.8 cdA	275.3 ± 3.5 bA	45.7 ± 3.0 cdeA	788.8 ± 2.0 cA	220.7 ± 3.0 bcA	117.5 ± 2.3 bcB	12.0 ± 0.1 abB	31.1 ± 0.0 abB	1849.8 ± 2.1 dA
I-DG3	288.1 ± 0.3 eB	177.2 ± 1.5 eB	35.0 ± 1.5 deA	673.6 ± 4.0 fB	142.0 ± 1.9 eB	101.4 ± 2.0 dB	13.0 ± 0.8 abB	23.7 ± 0.8 bcdB	1454.0 ± 5.7 hB
I-DB3	394.5 ± 3.6 abA	235.5 ± 0.6 cB	80.4 ± 0.4 aB	749.4 ± 9.4 dB	208.2 ± 1.5 cdB	108.3 ± 1.2 cdB	12.0 ± 1.4 abB	29.1 ± 2.0 abcdB	1817.4 ± 13.4 eB
I-DG6	370.0 ± 3.6 cA	263.1 ± 4.4 bB	40.5 ± 2.3 cdeA	815.2 ± 2.6 bB	217.7 ± 1.5 bcB	116.4 ± 0.3 bcB	9.5 ± 0.8 bB	33.7 ± 6.2 aB	1866.1 ± 2.2 dB
I-DB6	406.3 ± 7.3 aA	334.0 ± 0.6 aA	75.9 ± 0.2 abB	890.1 ± 3.7 aB	287.6 ± 0.3 aB	133.1 ± 0.1 abB	13.8 ± 0.1 abB	34.2 ± 0.2 aB	2175.0 ± 16.5 aB
I-DG9	297.5 ± 8.4 eA	269.5 ± 9.4 bB	32.8 ± 3.1 deA	713.5 ± 11.0 eB	222.4 ± 0.7 bcB	110.2 ± 2.5 cB	11.1 ± 2.1 abB	20.5 ± 1.4 dB	1677.5 ± 76.6 gB
I-DB9	387.7 ± 1.2 bA	267.6 ± 3.1 bB	66.0 ± 3.9 bA	792.7 ± 1.3 cB	233.9 ± 12.9 bB	110.8 ± 1.6 cB	12.5 ± 2.2 abB	32.1 ± 1.9 abB	1903.3 ± 6.9 cB
I-DG12	351.6 ± 0.5 dA	277.7 ± 2.0 bB	39.2 ± 2.8 cdeA	813.8 ± 3.2 bB	235.5 ± 1.0 bB	114.8 ± 1.6 cB	11.7 ± 1.8 abB	21.2 ± 1.1 cdB	1865.5 ± 54.3 dB
I-DB12	388.7 ± 2.8 bA	327.0 ± 3.7 aB	65.4 ± 2.5 bA	872.8 ± 5.8 aB	285.9 ± 12.6 aB	140.1 ± 11.5 aB	16.0 ± 4.7 abB	30.7 ± 1.4 abcB	2126.6 ± 7.5 aB
I-DG15	303.1 ± 1.9 eA	225.3 ± 1.0 dB	38.7 ± 5.3 eA	670.5 ± 0.2 daB	187.2 ± 0.7 dB	102.5 ± 3.4 dB	14.0 ± 0.6 abB	22.8 ± 1.7 bcdB	1564.1 ± 32.7 fB
I-DB15	351.3 ± 6.0 dA	337.7 ± 1.4 aB	50.0 ± 1.6 cB	877.6 ± 16.4 fB	291.7 ± 10.2 aB	140.3 ± 5.9 aB	18.8 ± 3.5 aB	26.1 ± 3.4 abcdB	2093.5 ± 6.6 bB
C-WG	251.3 ± 1.4 dB	216.7 ± 1.5 dB	32.5 ± 0.1 eB	689.0 ± 3.1 eB	233.8 ± 0.6 eA	186.6 ± 0.2 hA	27.7 ± 1.9 eA	42.5 ± 1.0 cA	1680.1 ± 3.0 gB
C-DG3	308.6 ± 1.8 bA	221.8 ± 3.3 dA	39.6 ± 1.1 dA	792.8 ± 8.1 dA	229.2 ± 3.7 eA	176.1 ± 3.0 hA	22.1 ± 2.9 eA	48.4 ± 0.9 bcA	1838.6 ± 7.2 fA
C-DB3	349.2 ± 0.6 aB	304.5 ± 4.5 cA	90.4 ± 1.2 aA	879.5 ± 0.2 cA	335.6 ± 1.0 dA	251.5 ± 0.1 dA	43.4 ± 2.1 bcdA	52.7 ± 1.7 abcA	2306.8 ± 20.7 bA
C-DG6	333.0 ± 6.5 bB	294.4 ± 1.8 cA	39.9 ± 1.1 dA	881.3 ± 3.7 cA	315.7 ± 1.6 dA	216.1 ± 0.9 fgA	35.2 ± 0.1 deA	58.0 ± 4.8 abA	2173.6 ± 27.1 dA
C-DB6	364.4 ± 3.2 aB	335.4 ± 4.1 bA	82.8 ± 0.9 bA	983.5 ± 9.0 aA	382.0 ± 0.6 bA	289.2 ± 2.1 bA	44.7 ± 1.1 abcA	55.9 ± 5.2 abcA	2537.9 ± 11.7 aA
C-DG9	290.2 ± 1.3 cA	290.6 ± 2.0 cA	33.0 ± 0.2 eA	818.6 ± 14.6 dA	321.5 ± 6.0 dA	211.3 ± 4.6 gA	39.8 ± 2.7 cdA	50.4 ± 1.0 bcA	2055.4 ± 57.4 eA
C-DB9	324.7 ± 0.4 bB	330.0 ± 10.3 bA	63.6 ± 1.8 cA	886.3 ± 20.8 bcA	365.2 ± 10.0 cA	269.6 ± 11.2 cA	36.0 ± 1.7 dA	55.2 ± 3.5 abcA	2330.6 ± 34.6 bA
C-DG12	315.4 ± 8.6 bB	316.8 ± 7.1 bcA	40.4 ± 1.6 dA	879.7 ± 7.9 cA	349.2 ± 4.2 cdA	232.2 ± 2.7 efA	42.6 ± 0.9 bcA	48.8 ± 7.1 bcA	2225.1 ± 15.7 cA
C-DB12	349.4 ± 7.3 aB	363.2 ± 4.3 aA	63.1 ± 0.9 cA	973.0 ± 6.6 aA	405.5 ± 5.2 aA	299.9 ± 0.9 abA	51.6 ± 1.1 aA	61.9 ± 3.3 abA	2567.6 ± 34.2 aA
C-DG15	314.5 ± 5.8 bA	302.2 ± 0.3 cA	36.5 ± 0.2 deA	923.2 ± 2.1 bA	360.7 ± 0.5 cA	243.7 ± 4.3 deA	43.6 ± 0.4 bcA	51.1 ± 3.8 bcA	2275.5 ± 41.3 cA
C-DB15	355.9 ± 3.2 aA	367.9 ± 13.4 aA	62.5 ± 0.9 cA	1002.0 ± 5.4 aA	419.0 ± 9.6 aA	310.4 ± 0.3 aA	48.7 ± 3.2 abA	67.0 ± 0.7 aA	2633.4 ± 3.4 aA

Abbreviation: I: Italian; C: Canadian; DG: debranned grain; DB: debranning by-product. Data are expressed as means ± standard deviation (*n* = 4). In each column, different lowercase letters mean significant differences (*p* < 0.05) among the fractions of each individual durum wheat variety (Italian and Canadian), whereas different capital letters mean significant differences (*p* < 0.05) between the same Italian and Canadian durum wheat fraction for each sterol.

## Data Availability

Data is contained within the article or Appendix A.

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
