# Peer review of "Lipid Process Markers of Durum Wheat Debranning Fractions"

_foods, 2023, doi:10.3390/foods12163036_

Round 1

Reviewer 1 Report

This research characterised the lipid components of different fractions obtained from the debranning of two kindes of durum wheat kernels with different percentages. It is meaningful for wheat flour industry. 

1. ABSTRACT: Some fatty acids, tocopherols and sterols could be useful biomarkers for evaluating the grain to tissue ratio in recombined flour. please clearly point out the specific fatty acids, tocopherols and sterols.

2. The tables showed lots of datas that are not intuitive enough, I suggest change them to figures.

3. The results are not deeply discussed, such as the section 3.3 and 3.4.

4. The conclustions should be improved.

It is ok for the quality of English.

Reviewer 2 Report

This is an interesting manuscript given the potential beneficial effects of wheat bran. The results appear to be solid. However, a major revision is needed to improve the clarity of method description and also to make this paper more valuable for the scientific community.

1. How was the total lipid content determined? This is not described.

2. Origin of α-tocopherol not given.

3. Origin of dihydrocholesterol not given. Is this the IS? 

4. How were the tocochromanol compounds identified? Do you have individual standards?

5. Are the phytosterol standards commercially available?

6. Can you show a chromatogram of tocochromanols? This is useful for readers who want to use your method.

7. Also can you supply a chromatogram of the phytosterol compounds?

8. The statistical symbols in Table 2 are quite confusing. Can you improve the footnote description?

9. Can you briefly describe the saponification process?

10. The conclusions regarding the lipid markers are not clearly described and justified.

Reviewer 3 Report

The main question addressed by the research is the assessment of the share of milling fraction in samples of recombinant whole grain flour. Recombination of milling fraction to obtained whole grains results in products with different properties. It is very useful to identify and use markers capable of distinguishing specific fraction used in the recombination prosess.

Round 2

Reviewer 1 Report

This paper can be accepted in present form.

Reviewer 2 Report

No further comments.